

**Diagnosing Ozone-NOx-VOCs-Aerosols Sensitivity to**
**Uncover Urban-nonurban Discrepancies in Shandong,**
**China using Transformer-based High-resolution Air**
**Pollution Estimations**
*Chenliang Tao[1], Yanbo Peng[1,2,*], Qingzhu Zhang[1,*], Yuqiang Zhang[1], Bing*
*Gong[3], Qiao Wang[1], Wenxing Wang[1]*
[1]Big Data Research Center for Ecology and Environment, Environmental Research
Institute, Shandong University, Qingdao 266237, P.R. China
[2]Shandong Academy for Environmental Planning, Jinan 250101, P. R. China
[3]Jülich Supercomputing Centre, Forschungszentrum Jülich, 52425 Jülich, Germany
**Keywords:**
Air pollution, Deep learning, Transformer, Satellite, Urban-rural difference, Ozone Regime
________________________________________________________________
*Corresponding authors. E-mail: zqz@sdu.edu.cn, pengyanbo@mail.sdu.edu.cn



**Abstract**
Narrowing surface ozone disparities between urban and nonurban areas escalate health
risks in densely populated urban zones. A comprehensive understanding of the impact
of ozone photochemistry processes on this transition remains constrained by our
knowledge of aerosol effects and the spatial availability of surface monitoring. Here we
developed a novel deep learning framework, which could perceive spatiotemporal
dynamics from adjacent grids by multidimensional self-attention operation, integrating
multi-sources data to estimate daily 500 m surface ozone, nitrogen dioxide ($NO_2$) and
fine particulate matter ($PM_{2.5}$) concentrations. Subsequently, three distinct ozone
formation regimes linked with its precursors, aerosols, and meteorology were
delineated through an interpretable machine learning method. The evaluations of the
framework exhibited average out-of-sample cross-validation coefficient of
determination of 0.96, 0.92 and 0.95 for ozone, $NO_2$ and $PM_{2.5}$, respectively. In 2020,
urban ozone levels in Shandong surpassed those in nonurban due to a more pronounced
decrease in ozone in the latter where $PM_{2.5}$ is the dominant anthropogenic driver. The
ozone sensitivity to volatile organic compounds (VOCs), the dominant regime in urban
areas, was observed to shift towards a $NO_x$-limited when extended to rural areas. A third
'aerosol-inhibited' regime was identified in the Jiaodong Peninsula, where the uptake
of hydroperoxyl radicals onto aerosols suppressed ozone production under low $NO_x$
levels during summertime. The reduction of $PM_{2.5}$ would increase the sensitivity of
ozone to VOCs, necessitating more stringent VOC emission abatement for urban ozone
mitigation. Our case study demonstrates the critical need for advanced modeling
approaches providing finer spatially resolved estimations.



## 1. INTRODUCTION


Surface ozone ($O_3$), fine particulate matter ($PM_{2.5}$) and nitrogen dioxide ($NO_2$) are
among the most important trace gases in the atmosphere that significantly impact the
ecological environment and public health (Han and Naeher, 2006; Yue et al., 2017).
During the Action Plan on the Prevention and Control of Air Pollution (denoted as the
Clean Air Action, 2013-2017) (Action Plan on Air Pollution Prevention and Control (in
Chinese), 2023), $PM_{2.5}$ and nitrogen oxide ($NO_x$ = nitric oxide (NO) + $NO_2$) emissions
across China decreased by 33% and 21% respectively (Zheng et al., 2018), while
surface $O_3$ exhibited an increasing trend (Lu et al., 2018). The increase in $O_3$ could be
partially attributed to the "aerosol-inhibited" effect, where the reduction in $PM_{2.5}$ results
in a diminished reactive uptake of hydroperoxyl radicals ($HO_2$) onto aerosol (Ivatt et
al., 2022; Li et al., 2019). The societal benefits of reducing premature deaths and
economic losses from $PM_{2.5}$ reductions have been diminished by the rising $O_3$ (Liu et
al., 2022). Thus, achieving the joint attainment objectives for $PM_{2.5}$ and $O_3$ has been put
on the top priority of China's long-term air pollution control policies.
The complexity of the $O_3$ formation is partly reflected by the nonlinear response
to changes in precursors (i.e. volatile organic compounds (VOCs) and $NO_x$), as well as
the presence of heterogeneous reactions in aerosols. Understanding these dynamics is
crucial to investigate the narrowing differences in $O_3$ concentrations between urban and
nonurban areas, which have traditionally shown higher levels in rural (Han et al., 2023).
The formaldehyde-to-$NO_2$ ratio (HCHO/$NO_2$ or FNR) serves as a theoretical gauge of



the relative abundance of total organic reactivity to hydroxyl radicals (OH) and $NO_x$
(Wei et al., 2022c; Sillman, 1995), and as such, it can function as a useful indicator of
$O_3$ sensitivity. Previous studies have utilized the $HCHO/NO_2$ from satellite remote
sensing to infer $O_3$ production regimes for guiding $O_3$ control policies (Jin et al., 2023;
Li et al., 2021a; Jin et al., 2020). However, the changes of $HCHO/NO_2$ threshold in $O_3$
regimes classification modulated by meteorology and localized atmospheric chemistry
in space and time, and uncertainties relating column to surface, precluding robust
applications over larger spatial scales (Lee et al., 2023; Jin et al., 2017; Souri et al.,
2023). While the observation-based model method alleviates some of these limitations,
constraints remain including computational demands and priori chemical mechanisms
(Song et al., 2022b; Chu et al., 2023). The advent of interpretable machine learning
models affords new opportunities to unravel intricate dependencies governing $O_3$
formation purely from actual observational data. However, sparse ground-based
monitoring stations, especially in rural areas, pose great challenges to the spatially full
coverage of studies. Thus, the high-spatiotemporal-resolutions estimations of surface
air pollutants are urgently needed to improve our understanding of how these pollutants
are changing and interacting.

Recent studies have utilized spatially resolved remote sensing data to estimate the

continuous distribution of air pollutants in space by diverse machine learning (ML)
models (Lyapustin and Wang, 2022; Lamsal et al., 2022; Huang et al., 2021; Li and
Wu, 2021; Ren et al., 2022b), such as random forest (RF), full residual deep learning,



and Bayesian ensemble model. These attempts have demonstrated the tremendous
potential of machine learning as an alternative to atmospheric chemical models (Jung
et al., 2022). Nevertheless, there are still several aspects that have not been fully
considered. For instance, coarse-resolution maps limit the ability to characterize the
fine-scale variation of air pollution within urban areas, which has significant
implications for environmental justice disparities of disadvantaged communities
(Jerrett et al., 2005; Ren et al., 2022b; Dias and Tchepel, 2018). Additionally, existing
machine learning models may not fully account for the complex atmospheric chemistry
and physics processes that influence pollutant concentrations due to the single-pixel-
based processing mode (Huang et al., 2021; Requia et al., 2020; Thongthammachart et
al., 2022; Li et al., 2022b; Geng et al., 2021). Although several efforts have been made
by using the neural network with convolutional layers (Di et al., 2016), and explicitly
incorporating spatiotemporally weighted information to machine learning models (Wei
et al., 2022b), the global spatio-temporal self-correlation of multi-dimensional features
in the input array remained unaddressed. Meanwhile, the convolutional operations
extract features from all neighboring grids of the target, ignoring the fact that the
environmental knowledge of the target grid itself is the most significant, with the
adjacent features being secondary.

Here, we developed a new spatiotemporal Transformer framework built

exclusively on self-attention over space, time, and variables, termed Air Transformer
(AiT), to reconstruct high spatiotemporal resolutions (daily, 500 m) estimations of





$PM_{2.5}$, $O_3$, and $NO_2$ from TROPOMI. In this framework, we paid special attention to air
mass transport and dispersion affected by the spatial-temporal correlations,
incorporated the downscaling mechanism from the model perspective, and considered
the interactions between multiple pollutants from massive ground-level monitoring,
satellite observations, meteorological conditions, dynamic industrial emissions, and
other ancillary data. The explainable method (Shapley Additive exPlanations, SHAP)
(Lundberg and Lee, 2017) was leveraged to provide insights into the impact of each
environmental factor on air quality. The fidelity of the dataset was evaluated by
comparing the spatial-temporal variations of RF (Breiman, 2001) estimations with the
same variables and also the ChinaHighAirPollutants (CHAP) dataset (Wei et al., 2022b,
2020, 2022a). The spatial characteristics of air pollution from various emission sources
and the urban-nonurban disparities across different cities are further examined to
elucidate the potential values of high-resolution data. Surface $O_3$ formation regimes in
Shandong provinces were inferred by the classic XGBoost model (Chen and Guestrin,
2016) coupled with SHAP, which identifies the impact of meteorology, $PM_{2.5}$, $NO_2$ and
HCHO on $O_3$, in which HCHO was derived using the conversion factors algorithm
based on the Tropospheric Ozone Monitoring Instrument (TROPOMI) and reanalysis
of atmospheric composition. The new deep learning framework is expected to enable
new applications like those of fine-scale air quality simulation, health exposure
assessment, and $O_3$ formation regimes studies.





**2. MATERIALS AND METHODS**
**2.1 Predictor Variables**
The study domain covered the Shandong provinces of China with a high mortality
burden of air pollution (Liu et al., 2017). The surface $PM_{2.5}$, $O_3$, and $NO_2$ concentration
measurements were collected from the regulatory air quality stations of the China
National Environmental Monitoring Center (CNEMC, with a total of 179 locations)
and also the Shandong Provincial Eco-environmental Monitoring Center (SDEM, with
a total of 166 locations) (Figure S1). The SDEM stations were included to fill the spatial
gap in the county and rural areas where CNEMC stations were lacking. The study area
was divided into 1.22 million grid cells with a spatial resolution of 500 m. We utilized
a range of predictor data including tropospheric $NO_2$ vertical column densities (VCDs)
and $O_3$ total VCDs measured by TROPOMI (Lamsal et al., 2022, 2020), aerosol optical
depth (AOD) data and atmospheric properties obtained from Moderate Resolution
Imaging Spectroradiometer (MODIS) Multi-Angle Implementation of Atmospheric
Correction products (Lyapustin and Wang, 2022), AOD estimates from Modern-Era
Retrospective Analysis for Research and Applications as the supplement of MODIS
(2015), meteorological reanalysis data obtained from ERA5 (Hersbach et al., 2023, p.5),
daily dynamic industrial emissions, moonlight-adjusted nighttime lights product
(Román et al., 2018), vegetation index (Didan, 2021), population density (WorldPop,
2018), road density, land use data (Jun et al., 2014), and the shuttle radar topography
mission digital elevation model. The detailed information for all the predictive variables





is listed in Table S1 and discussed in Text S1-2. Taking the space-variant and seasonal
patterns into consideration, several spatiotemporal indicators such as geographical
coordinates, Euclidean spherical coordinates, year, Julian date, and helix-shape
trigonometric sequences were also included as predictor variables (Text S3) (Sun et al.,
2022). Geographic Information Systems techniques including reprojection and
resampling were used to consolidate all the data obtained for consistent projection and
spatial scale. Finally, the Light Gradient Boosting Machine was used to fill satellite data
gaps (Text S4) (Ke et al., 2017).
**2.2 Air Transformer**
AiT is an individual Transformer model that adopts encoder-decoder architecture
with multidimensional self-attention computation to dynamically capture the
spatiotemporal autocorrelation of atmospheric pollution changes from the sequences of
pixels and variables for more reliable spatial maps of estimation. Compared with the
existing image and video recognition Transformers, such as ViT (Dosovitskiy et al.,
2021), Timesformer (Bertasius et al., 2021) and Uniformer (Li et al., 2021b), AiT is
innovative in incorporating self-attention across channels after the self-attention based
on pixels and taking advantage of the decoder. The former can capture the correlations
between predictor variables. The decoder was employed to enable interaction between
the primary target grid and neighboring grids. Predictor variables with 8-timesteps
within 1000 m of the target grid cell were fed into the model to learn spatiotemporally
disparities among atmospheric pollutants for predicting $O_3$, $NO_2$ and $PM_{2.5}$ within the



target grid point.

The overall architecture of the proposed AiT model and the dimension of input

data are illustrated in **Figure 1**. The encoder maps an input sequence with neighborhood
spatiotemporal data to a sequence with high-dimensional spatial-temporal
characteristics, and the decoder generates an estimation by computing self-attention
representations between the target grid and outputs of the encoder. The encoder of AiT
takes as input a clip $X \in R^{V \times T \times H \times W}$ consisting of $T$ multi-variables frames of size
$H \times W$ sampled from the original dataset, where $V$ is the number of variables and the
target grid cell is located in $\left(\left\lceil \frac{H}{2} \right\rceil, \left\lceil \frac{W}{2} \right\rceil\right)$. The decoder takes as input a clip $X \in R^{V \times 1 \times 1 \times 1}$
consisting of $V$ variables from the target grid. Several Transformer blocks with
modified self-attention computation (AiT blocks) are applied to the encoder. The AiT
encoder block is similar to the standard vision transformer block but specifically
designed for atmosphere estimation (Dosovitskiy et al., 2021). It is a stack of two self-
attention schemes including global spatiotemporal self-attention on the pixels and
channel self-attention on variable predictors. The former contains $N = HW$ effective
input sequence length for the self-attention to extract spatiotemporal information. The
latter computes self-attention based on $V$ effective input sequence length to capture
hidden variables information. The decoder part is symmetric to the encoder part, while
it only has a block with the spatiotemporal self-attention mechanism. We compute the
matrix of self-attention outputs as:

$$Attention(Q, K, V) = \text{softmax}\left(\frac{QK^T}{\sqrt{d_k}} + B\right)V \qquad (1)$$



where $Q$, $K$, and $V$ are the queries, keys, and values are the inputs of the particular
attention, respectively. $d_k$ is the feature dimensionality of the $K$, and $B$ is the
geographic positional bias term. Another difference is that the attention function of the
decoder is computed on $Q$ from the estimated grid data, and $(K, V)$ from the outputs
of encoder blocks under the same stage, resulting in the outputs of the last decoder
block sized $1 \times 128$. The description of the data transformation and design details in
the process of training can be found in Text S5. The multi-task learning strategy was
also applied for learning representation across multiple pollutant estimation tasks (Text
S6).

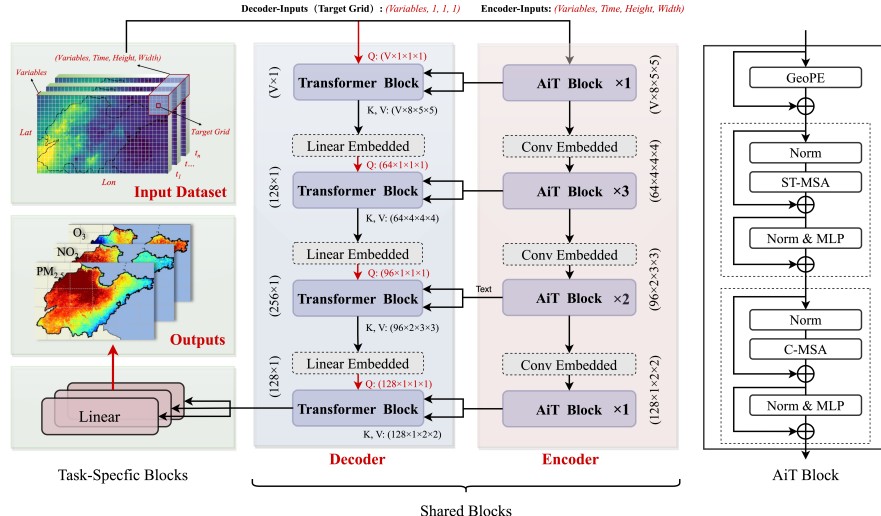


**Figure 1.** Schematic diagram of the AiT model. The white box of multi-dimension
inputs presents each pixel of raster data. The AiT Block is a Transformer block based
on self-attention across space, time and variables. The GeoPE, Norm, MLP, ST-MSA
and C-MSA indicate respectively positional embedding, layer normalization, multi-
layer perceptron, spatial-temporal multi-head self-attention and multi-channels (multi-
variables) multi-head self-attention.



**2.3 Diagnosing O$_3$ Formation Sensitivity**

Interpretability can provide insight into how a model may be improved, bolster understanding of the process being modeled, and engender appropriate confidence from researchers. SHAP is a coalitional game theoretic approach based on Shapley values (Shapley, 1988) and then assigns each variable an importance value for a particular estimation. Deep SHAP, a high-speed approximation algorithm that builds on the connection between Shapely values and DeepLIFT (Shrikumar et al., 2019), is employed to compute the feature importance of AiT from all data with monitoring labels for interpreting the prediction. The sensitivity of the O$_3$ formation regime was deduced using a combination of the XGBoost model and SHAP interpretability method using the GPUTreeShap algorithm (Mitchell et al., 2020), which simulated the response of surface O$_3$ to meteorological conditions, HCHO, NO$_2$ and PM$_{2.5}$, by utilizing the continuous estimations from ERA5, AiT and TROPOMI between 2019 and 2020. The incorporation of meteorology in the model ameliorated the inadequacies in the conventional method (HCHO-NO$_2$ ratio) where its thresholds for identifying O$_3$ regimes vary temporally and spatially. The positive or negative contributions of three atmospheric pollutants were used to identify their promoting or inhibitory effect on O$_3$ variability. Given the unbiased property of SHAP values regarding directionality, the normalized relative magnitudes of SHAP values were calculated for HCHO (a proxy for VOCs), NO$_2$ (a proxy for NO$_x$) and PM$_{2.5}$ (a proxy for aerosols). This allowed differentiation of the O$_3$ formation regimes based on the locally maximal proportions





of the SHAP values for each species. The ground-level monthly HCHO concentrations
were derived using a combination of column-to-surface conversion factor (CF)
simulated from the ECMWF Atmospheric Composition Reanalysis 4 and the
tropospheric HCHO VCDs obtained from the TROPOMI (Cooper et al., 2022; Su et al.,
2022; Inness et al., 2019). A detailed description of the CF method as used here is
discussed in Text S7. To ensure consistency in resolution between TROPOMI and AiT,
we employed the oversampling method to downscale the TROPOMI VCDs to the
resolution of AiT estimation, which has been proven to be effective in achieving finer
resolution (Su et al., 2022; Cooper et al., 2022; van Donkelaar et al., 2015).

### 237    3. RESULTS AND DISCUSSION

### 238    3.1 Performance Evaluation for the AiT

### 239    3.1.1 Cross-validation Metrics

We evaluated the AiT performance based on the 10-fold cross-validation (CV)
approach (Text S8), with correlation coefficient ($R^2$) measuring the extent to which
model simulations explain variability in atmospheric pollutants, and root mean square
errors (RMSE) and mean absolute errors (MAE) evaluating the bias/error of the
estimates. As shown in **Figure 2**, out-of-sample CV daily ground-level $O_3$, $NO_2$ and
$PM_{2.5}$ estimations are highly consistent with ground observations ($R^2$ = 0.96, 0.92, 0.95),
indicating low uncertainties, with RMSE of 10.1, 4.7, and 8.5 $\mu g/m^3$ and MAE of 7.2,
3.5, and 5.3 $\mu g/m^3$ during the 2018-2021 period. The linear regression comparing the
$O_3$ predictions versus observations yields a slope of 0.98 and an interception of 2.39,



which demonstrates that there is no systematic bias in the estimations. Meanwhile, as

shown in Figure S2, our AiT model works well in the individual-site scale with high

CV-RMSE for $O_3$, $NO_2$, and $PM_{2.5}$ (10.5 ± 8.6, 4.7 ± 1.1, and 8.3 ± 2.8 µg/m$^3$). In

general, AiT model is robust for multi-pollutant simultaneous estimations.

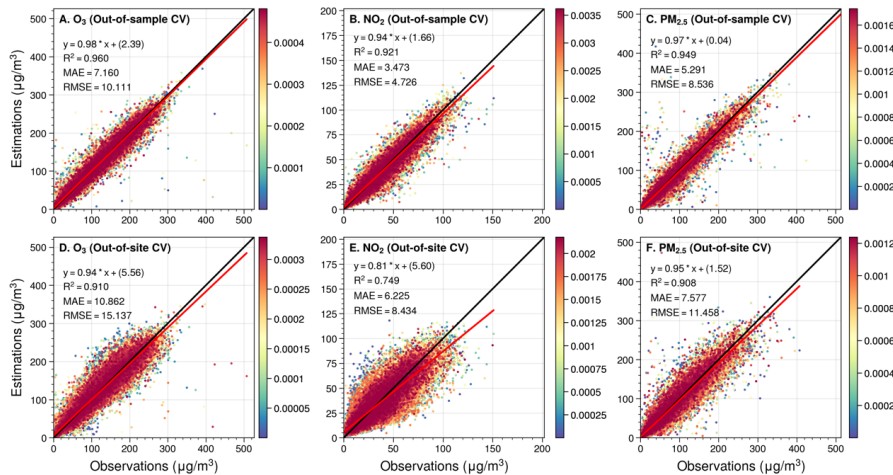

**Figure 2.** Out-of-sample cross-validation (A-C) and out-of-site cross-validation (D-F)
of daily ground-level $O_3$, $NO_2$ and $PM_{2.5}$ concentration in the validation set.

The spatial generalization ability of AiT is then examined by the out-of-site CV

evaluation method (**Figure 2**). The daily spatial variations of $O_3$, $NO_2$, and $PM_{2.5}$ at

locations where there are no ground measurements can be well estimated by our model

(i.e., CV-$R^2$ = 0.91, 0.75, 0.91), which is a core contribution of such studies. We also

probe the model performance for each site separately based on spatial CV estimations

(Figure S3). This general model yields RMSE of 15.2 ± 8.8, 8.1 ± 2.7, and 11.1 ± 2.8

µg/m$^3$, respectively. Furthermore, we trained AiT model using data exclusively from

CNEMC and assessed its generalizability by validating it with data from SDEM. The



model demonstrates strong performance with high $R^2$ values in the validation dataset
of CNEMC (Figure S4), and when evaluated with SDEM data, it exhibits only a slight
degradation in predictive accuracy ($R^2$ for $O_3$, $NO_2$, and $PM_{2.5}$: 0.95, 0.89, 0.85).
Meanwhile, our framework utilizes multi-task learning to enhance computational
efficiency through a single iteration and leverages the interactions among multiple
pollutants to optimize the performance of individual pollutant levels (Table S2). In
summary, AiT provides relatively stable estimations in areas without available ground-
level monitoring values and reliably extends ground monitoring from the site scale to
the full-coverage spatial scale with high spatial resolution.
**3.1.2 Compared with Other ML Models**
Since the ground-level air quality measurements across the target regions are
extremely limited at 500 m spatial resolution, representing only roughly two-
thousandths of the total grid cells, we seek implicit approaches to validate our estimated
near surface pollutants concentrations. We compared the model performance with
previous studies that applied different ML methods to estimate these three air pollutants
individually and found out that our cross-validation results are comparable or even
better than those (Table S3). We also created a new dataset in our study by applying the
classic RF algorithm which is the most common ML model for estimating atmospheric
pollution in recent years (Wei et al., 2022a; Requia et al., 2020; Xiao et al., 2018; Geng
et al., 2021; Lu et al., 2021) with the same variables as AiT. The statistics comparisons
between AiT and RF are also shown in Table S3. We then compared the spatial





distribution of our results with estimations from CHAP.
**Figure 3** shows the spatial maps of near-surface air pollutants with partially
zoomed satellite images for monitoring sites, AiT, RF and CHAP in 2019 (see Figure
S6 for 2020). We found that the estimated $NO_2$ and $PM_{2.5}$ from the AiT share a similar
spatial distribution as those estimated by RF and CHAP. However, enlarged city-level
urban regions in **Figure 3** reveal that AiT estimates fine structures and intra-urban
disparities in near-surface multi-pollutant concentrations which cannot be captured by
either RF or CHAP products. In general, while RF and CHAP can only see the hotspots
of air pollutants at a regional scale, the spatial distribution of air pollutants estimated
by the AiT shows much more detailed differences with high spatial and temporal
variability across the city scale. The difference of near-surface annual averaged
pollutants between 2019 and 2020 for measured and multi-estimated data were
presented in Figure S7. The reductions or increments of $O_3$, $NO_2$ and $PM_{2.5}$ in distinct
locations can be simulated by our model, which is relatively consistent with the change
of measurements. The zoomed maps of Figure S7 show the difference in three pollutant
concentrations at the city scale of the capital of Shandong Province, Jinan. It can be
found that the change in pollutant levels in 2020 compared to 2019 exhibits substantial
regional variations and intra-urban heterogeneity, with some areas experiencing an
increase while others a decrease. Compared to estimations of RF and CHAP, our results
successfully capture the complex distribution of air pollution in reality and reveal that
the decline of $PM_{2.5}$ is primarily concentrated in suburban areas, while an increase is





pronounced in some regions of urban during 2020. Notably, this spatial trend may be
consistent with the underlying emission patterns and meteorological conditions.

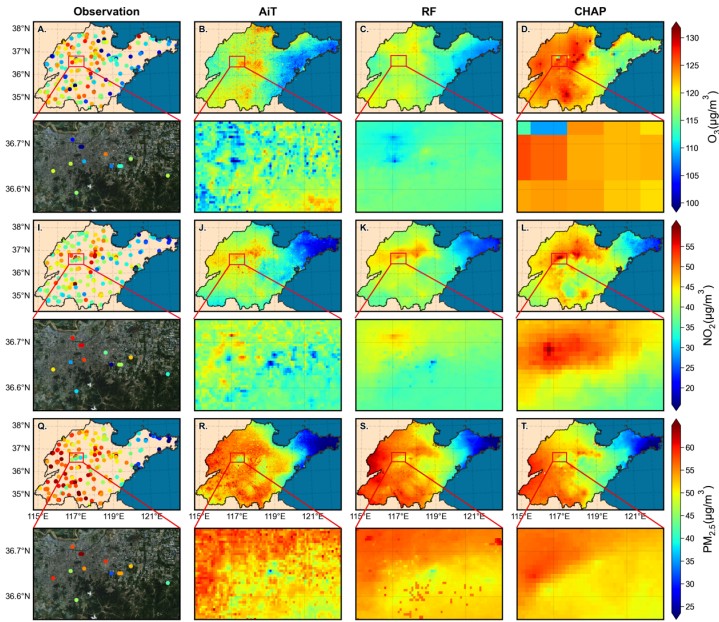


**Figure 3.** Spatial distribution of the annual mean (A-D) $O_3$, (I-L) $NO_2$ and (Q-T) $PM_{2.5}$
concentrations from observations, Air Transformer (AiT), Random Forest (RF) and
ChinaHighAirPollutants (CHAP), respectively, in 2019. The region enclosed by the red
rectangular box in (A-T) corresponds to the zoomed-in maps of satellite (© Tianditu:
www.tianditu.gov.cn) and pollutant concentrations at a city scale for the capital city of
Shandong Province, Jinan

**3.1.3 Typical Event Study**

The typical example of the spatial distribution of multi-pollutant observations and

estimations of AiT is compared for validating the predictive capability of the model at
a particular pollution episode, i.e., 13-16 March 2021. During this period, an early
season dust storm, which was called the largest and strongest such storm in a decade,



hits northern China (Myers, 2021). As shown in Figure 4, our model can capture the
spatial distribution of surface $O_3$, $NO_2$ and $PM_{2.5}$ in the time of severe atmospheric
pollution. In addition, our estimations are highly concordance with measurements in
terms of magnitudes and spatial variability over the entire research region. Combined
wind fields to analyze $PM_{2.5}$ distribution on the day of the dust storm, it can be found
that surface wind carries a massive amount of particulate matter from Beijing, which
suffered a severe dust storm, to northern Shandong. The influence was gradually
diminishing in southern Shandong due to the obstruction of Mount Tai. Spatial
heterogeneity within intra-urban was further investigated to identify the hotspots of
pollution sources. The satellite images in even-numbered rows of Figure 4 illustrate the
spatial disparities of three pollutants around four typical emission sources: thermal
power plants, industrial parks, overpasses and parks. As depicted, these anthropogenic
emission sources contribute to higher pollution levels, while the mountain in the park
mitigates primary pollution but increases $O_3$ concentrations. Industrial sources emit a
large number of $NO_x$ and $PM_{2.5}$, leading to increased pollution of these species
compared with other urban microenvironments, which in turn promotes $O_3$ formation,
particularly in downwind areas (Miller et al., 1978; Tang et al., 2020). Although the
spatial gradients of pollutants on the street are not as apparent as in the dataset with 100
m resolution (Huang et al., 2021), the predicted spatial variation between various
geographical scenes is in satisfactory agreement given the 500 m scale of the model.
Urban areas affected by diverse dust pollution exhibit lower $PM_{2.5}$ concentrations



compared to rural due to the obstructive and filtering effects of artificial structures such
as buildings and urban greenery (Figure S8), which cannot be effectively captured
solely by ground-based observations. Notably, the elevated $PM_{2.5}$ inhibits the formation
of $O_3$ by diminishing solar radiation flux and absorbing the $HO_2$ radical on the aerosol
surface, even in conditions characterized by similar $NO_2$ levels. As the mapping, AiT
accurately grasps the spatial characteristic of air pollutants and delivers a coherent
spatial-temporal distribution that is consistent with the prior knowledge of atmospheric
transport.

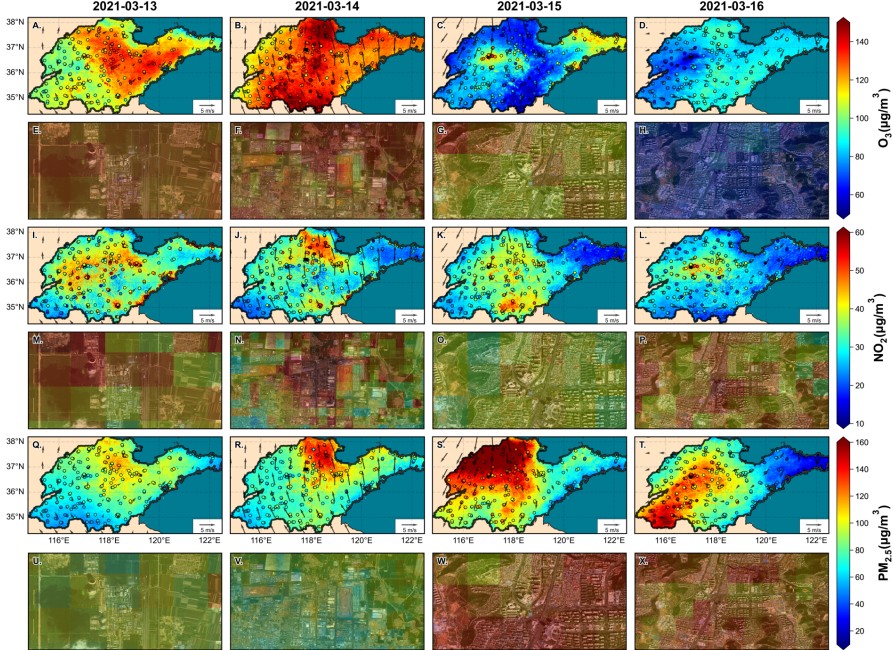


**Figure 4.** The spatial distribution of ground-level $O_3$ (A-D), $NO_2$ (I-L), and $PM_{2.5}$ (Q-
T) concentrations from AiT and monitoring stations during 13-16 March 2021 in
Shandong, China. The black arrows are the 10 m wind speed and wind direction. The
even-numbered rows correspond to the concentration distribution maps of typical
emission sources for the respective pollutants, accompanied by satellite images (©
Tianditu: www.tianditu.gov.cn). The upper right area of E, I, and Q is a thermal power



plant in Weifang City (119°250′E-119°280′E, 36°658′N-36°673′N). The center area of
F, N, and V is an industrial park in Zibo city (117°725′E-117°845′E, 36°880′N-
36°940′N). The center and upper right area of G, O, and W is an overpass and Wanling
mountain in Jinan city (116°977′E-117°009′E, 36°590′N-36°606′N). The center area of
H, P, and X is another overpass in Jinan city (116°970′E-117°030′E, 36°580′N-
36°610′N).
**3.2 Urban-nonurban Difference**

The advantage of full-coverage pollutant estimates is the ability to assess the

difference between urban and non-urban areas on a finer scale. Table S4 shows the
concentrations of $O_3$, $NO_2$, $PM_{2.5}$ and HCHO over the urban and nonurban regions,
delineated from an annual urban extents dataset (Zhao et al., 2022). From 2019 to 2020,
surface air pollutant levels declined significantly in Shandong. The averaged
concentration discrepancies of these pollutants between urban and non-urban over
February to March (lockdown during COVID-19) and June to October (summertime)
as shown in **Figure 5**. Surface concentrations in $NO_2$ and HCHO are higher in urban
than nonurban areas, and the differences narrowed from February to October, while
$PM_{2.5}$ is opposite at both. The ground-level $O_3$ levels exhibited unexpected urban-
nonurban disparity variations, from the lockdown period through the summer, as well
as from 2019 to 2020. In comparison to nonurban areas, the urban, which previously
had lower $O_3$ levels, began to experience higher concentrations, attributed to a more
rapid decline of ozone in nonurban regions. **Figure 6** revealed that urban-nonurban
differences in $O_3$ and $PM_{2.5}$ varied across various cities during the lockdown period in
2019, while the higher $NO_2$ pollution in urban remained consistent. In summer, only a
handful of urban areas exhibit lower levels of ozone concentration, where $NO_2$ and



$PM_{2.5}$ levels surpass those in nonurban regions, attributable to a more pronounced
titration effect of NO and a slower rate of photochemistry reactions (Figure S9) (Sicard
et al., 2016, 2020; Zhang et al., 2004). Comparative urban-nonurban differences from
2019 to 2020 indicate an accelerated reduction of ozone and HCHO in non-urban areas,
while $NO_2$ and $PM_{2.5}$ levels in urban have seen a more significant decrease due to the
decline in anthropogenic activities, particularly the suspension of emissions from
pollution sources located in urban areas. Upon comparing the results of urban-nonurban
disparities of our data with monitoring data and the CHAP dataset, we have identified
potential overestimations or underestimations across various cities in monitoring data,
likely resulting from the limited number of non-urban sites (**Figure 6**M, S10). The
urban-nonurban difference calculated by the CHAP generally aligns with our findings
(Figure S11). Nevertheless, it is worth noting that the coarse resolution of $O_3$ (10 km)
has led to a significant overestimation. These results highlight the invaluable value of
high-resolution and gapless data for studying urban-nonurban disparities.

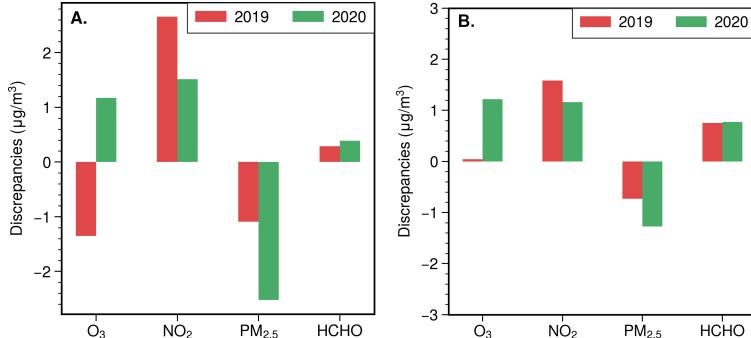


**Figure 5.** The discrepancies of $O_3$, $NO_2$ and $PM_{2.5}$ between urban and non-urban from
2019 to 2020 for the lockdown period (A) and summertime (B) averaged concentration.

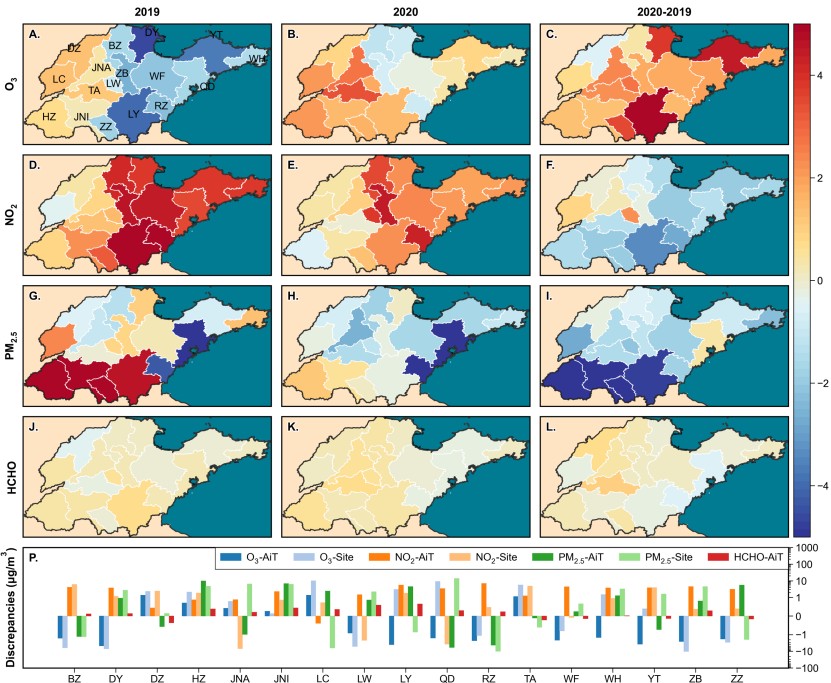

**Figure 6.** The urban-nonurban disparities of $O_3$, $NO_2$, $PM_{2.5}$ and HCHO calculated by AiT across cities with administrative divisions in Shandong, China during lockdown periods in 2019 (A, D, G) and 2020 (B, E, H), and the changes of differences between 2019 and 2020 (C, F, I). M is the comparison between the results of monitoring station data and AiT dataset in 2019. The red color represents a greater decline in air pollutants in nonurban areas, while the blue color indicates a more significant reduction in urban areas in the third column of the figure. (YT: Yantai, BZ: Binzhou, DY: Dongying, WH: Weihai, DZ: Dezhou, JNA: Jinan, QD: Qingdao, WF: Weifang, ZB: Zibo, LC: Liaocheng, LW: Laiwu, TA: Taian, LY: Linyi, RZ: Rizhao, JNI: Jining, HZ: Hezhe, ZZ: Zaozhuang)

**3.3 Photochemical Regimes**

**3.3.1 Ozone-NO$_x$-VOCs-Aerosols Sensitivity**

Figure S12 shows the seasonal maps of $O_3$, $PM_{2.5}$ and $NO_2$ estimations from AiT, satellite-derived surface HCHO. Based on these data, we first capture the well-established non-linearities in $O_3$-VOC-NO$_x$ chemistry by a conceptual framework



similar to classic $O_3$ isopleths typically generated with models (Pusede et al., 2015; Ren
et al., 2022a). **Figure 7**a depicts $O_3$ concentration as a function of HCHO and $NO_2$,
which was derived solely from ground-level estimation. The result indicates that the $O_3$
regimes can be qualitatively identified based on the nonlinear interaction between
surface $O_3$, HCHO and $NO_2$. In the regime characterized by high $NO_2$ and low HCHO,
the elevated consumption of $HO_x$, predominantly driven by the $OH + NO_2$ termination
reaction, results in the suppression of $NO_x$ on $O_3$, indicating the prevalence of VOC-
limited chemistry. Conversely, when HCHO levels are high and $NO_2$ levels are
relatively low, $O_3$ increases with $NO_2$ and exhibits insensitivity to HCHO due to
abundant peroxyl radicals ($HO_2$ + organic peroxy ($RO_2$) radical, $RO_x$) self-reactions,
suggesting a $NO_x$-limited (VOC-saturated) chemistry. In the case of high HCHO and
$NO_2$, the $O_3$ increases with both HCHO and $NO_2$, reaching a peak. While **Figure 7**a
resembles this overall $O_3$-VOC-$NO_x$, the blurry transition between two different
regimes and the role of $PM_{2.5}$ is uncertain which may be influenced by meteorological
conditions, chemical and depositional loss of $O_3$, errors of estimations, and "aerosol-
inhibited". Increased $PM_{2.5}$ levels could suppress $O_3$ formation even under high HCHO
and $NO_2$ conditions (**Figure 7**b), which could be induced by enhanced reactive uptake
of $HO_2$ onto aerosol particles and weaker photochemical reaction resulting from the
scattering and absorption of solar radiation by anthropogenic aerosols. The relationship
between $PM_{2.5}$ and $O_3$ in Shandong demonstrated the distinct stages of $O_3$ chemistry, as
depicted in **Figure 7**c. When $PM_{2.5}$ was below the maximum turning point (MTP1, 35



μg/m$^3$), a linear and positive correlation between $O_3$ and $PM_{2.5}$ was observed due to the
common dependence on their precursors in the initial stage (Zhang et al., 2022). As
$PM_{2.5}$ increased beyond the MTP1, a sharp reduction in HCHO and $O_3$ was observed,
accompanied by a decline in surface short-wave radiation, which reflect their formation
as a photo-oxidation product of OVOCs and $NO_x$. When $PM_{2.5}$ exceeded the minimum
transition point (MTP2, 45 μg/m$^3$), a phase was observed with stagnant radiation
intensity and relatively higher $NO_2$ levels compared to HCHO. This regime is typically
associated with a VOC-limited regime, where increased HCHO and decreased $NO_2$
concentration could promote $O_3$ production. However, our findings demonstrated an
opposite impact of HCHO and $NO_2$ on $O_3$ when $PM_{2.5}$ beyond MTP2. **Figure 7**d shows
the changes in the quantitative relationships between HCHO/$NO_2$ (FNR) and $O_3$ by
artificially changing $PM_{2.5}$ and precursors levels for XGBoost, in which the peak of
curves marks the transitional threshold of $O_3$ regimes from VOC to $NO_x$ sensitive. It
can be seen that attenuated $PM_{2.5}$ pollution could increase the sensitivity of $O_3$ to VOCs
and decrease the sensitivity to $NO_x$, which causes the shift in $O_3$ regimes from $NO_x$-
limited to VOC-limited. With the recent reduction in $NO_x$ emission in China, the
anticipated transition of $O_3$ production regime in urban areas towards more $NO_x$-limited
has been impeded by the heightened VOC sensitivity resulting from decreased $PM_{2.5}$
levels. Our results are consistent with the findings of Li et al. on the $O_x$-$NO_x$ relationship
in response to changing $PM_{2.5}$ (Li et al., 2022a), and the findings of Dyson et al. on the
impact of $HO_2$ aerosol uptake on $O_3$ production (Dyson et al., 2023). The SHAP





interaction values between $PM_{2.5}$ and the other two variables, HCHO and $NO_2$,
demonstrated that lower $NO_2$ and higher HCHO levels could diminish the formation of
$O_3$ under high $PM_{2.5}$ concentrations due to enhanced titration of $O_3$ by NO resulting
from weaker conversion from NO to $NO_x$ through $RO_x$ radical (**Figure 7**e, f). It further
illustrates that the scavenging of $HO_2$ on aerosols can cause the shift of $O_3$ regimes from
being VOC-limited to $NO_x$-limited and the threshold approach is restricted by aerosol
and meteorology for determining the constantly changing $O_3$ formation regimes over
time and space.

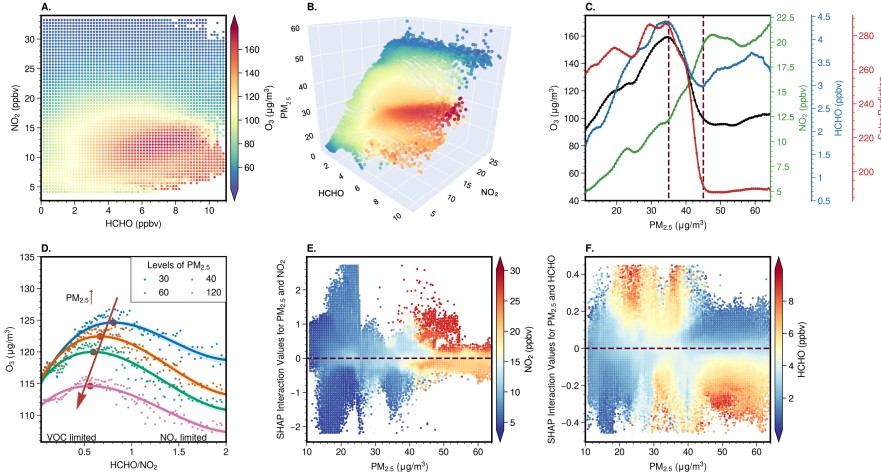


**Figure 7.** (A) $O_3$ concentrations as a function of surface HCHO and $NO_2$. (B) $O_3$
concentrations as a function of surface HCHO, $NO_2$ and $PM_{2.5}$. (C) Relationship
between $O_3$, and $NO_2$, HCHO and surface short-wave radiation flux. The paired $O_3$,
HCHO, $NO_2$ and solar radiation are divided into 100 bins based on $PM_{2.5}$ and then the
averaged concentrations (y-axis) are calculated for each $PM_{2.5}$ bin (x-axis). (D)
Changes in HCHO/$NO_2$-$O_3$ relationship in response to changing $PM_{2.5}$ by XGBoost
model. The solid lines are fitted with four-order polynomial curves, and the shading
indicates 95% confidence intervals. (E-F) The interaction SHAP values reveal an
interesting hidden relationship between pairwise variables ($PM_{2.5}$ and $NO_2$, HCHO) and
$O_3$.



Unraveling the intricate interplay of $O_3$ on meteorology, aerosol and precursors

that govern $O_3$ formation over extensive spatial domains has long confounded robust
interpretation. These multiscale processes were elucidated by using an interpretable ML
model, which can quantify the positive or negative contributions of individual processes.
Figure S13 elucidates that meteorological variations, chiefly surface short-wave
radiations flux modulating photochemical reaction kinetics, primarily dictate the
heterogeneous geographic distribution of $O_3$ at the regional scale, with lower levels
over Jiaodong Peninsula. Meanwhile, local atmospheric chemical processes
predominate the city-scale variability of $O_3$. HCHO facilitated $O_3$ formation in urban
areas yet suppressed it in rural regions across areas with high ozone, where most $NO_2$
promoted $O_3$ production overall, indicating VOC-$NO_x$ synergistic control on $O_3$ in cities
and a $NO_x$-limited regime in rural areas during summertime. The contribution of $NO_2$
and $PM_{2.5}$ exhibits analogous seasonal variability, promoting $O_3$ formation under low
pollution conditions while inhibiting $O_3$ when pollution levels are high (Figure S12,
14). The elevated $NO_2$ levels in autumn led to a negative contribution to $O_3$, whereas
the facilitating effect of $PM_{2.5}$ was enhanced. This stems from the relatively moderate
$PM_{2.5}$ concentrations slightly affecting photochemical reaction rates, while the
increased $NO_2$ amplified the reactive uptake of $NO_2$ by $PM_{2.5}$, generating more OH
radicals that promote $O_3$ formation (Lin et al., 2023; Tan et al., 2022). In winter, $PM_{2.5}$
pollution exceeding 75 $\mu g/m^3$ suppressed $O_3$ formation through scattering and
absorbing solar radiation that activates atmospheric chemical processes, which



counteracted the promoting effect of high $PM_{2.5}$ through the conversion of $NO_2$ to
HONO.

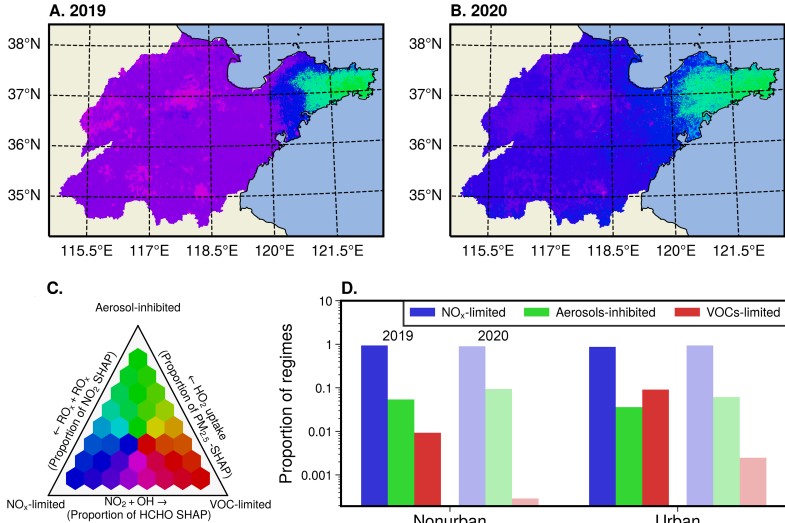


**Figure 8.** Comparison of geographical distribution for ozone formation regimes
between 2019 and 2020 in the summertime. All surface daily $O_3$, $PM_{2.5}$ and $NO_2$
estimations from Air Transformer (AiT) are averaged over each month from May to
October 2019-2020 for matching monthly HCHO derived from TROPOMI (500 * 500
m). (A, B) Geographical distribution of fractional contribution of chemical factors
representing $O_3$ formation regimes. (C) The ternary phase diagram depicts the
normalized fraction of SHAP values for $O_3$ attributed to HCHO, $NO_2$, and $PM_{2.5}$ at the
surface, representing VOC-limited (red), $NO_x$-limited (blue) and aerosol-inhibited
(green) regimes. (D) The proportion and changes of three regimes across urban and
nonurban areas in Shandong between 2019 and 2020.

     **Figure 8** shows the ternary phase diagram and surface distribution of the relative

proportion of SHAP values on three pollutants for inferring the photochemical regimes

of $O_3$. More of urban regions in Shandong are pink, indicating a VOC-limited regime

where $NO_2$ + OH is the dominant termination step, which is consistent with the findings

of previous studies on major cities (Ren et al., 2022a). Moving along an urban-to-rural





gradient, reactions dominated by $RO_x$ radical self-reactions continuously enhanced with
the increasing $NO_x$ SHAP values, resulting in the majority of rural Shandong being
situated in $NO_x$-limited regimes. Furthermore, the overall ozone production regimes in
Shandong exhibited a transition toward more $NO_x$-limited from 2019 to 2020, with
regions dominated by $NO_x$-limited shifting toward aerosol-inhibited in the Jiaodong
Peninsula. The aerosol-inhibited regime differs from either of the two classically
applied tropospheric $O_3$ policy-control regimes. It is attributed to predominant
heterogeneous $HO_2$ uptake by aqueous aerosols, despite comparatively low $PM_{2.5}$ levels
during summertime. The marine environment engenders liquid aerosol particles with
$HO_2$ uptake coefficients exceeding those of dry aerosols by orders of magnitude (Song
et al., 2022a). Concurrently, lower ambient $NO_x$ levels minimize the promotive effects
of aerosols on ozone formation (Tan et al., 2022; Kohno et al., 2022). This result is
consistent with the results of Dyson et al. (Dyson et al., 2023), which concluded that
the contribution of $HO_2$ sinks onto aerosols on total $HO_2$ could increase for areas with
low NO levels. The attenuated responsiveness of $O_3$ formation to VOCs induced by the
uptake of $HO_2$ results in enhanced sensitivity of $NO_x$ at the northwest boundary region
of the Jiaodong Peninsula. Collectively, these processes delineate an aerosol-inhibited
ozone production regime, reflecting the sensitivity of $O_3$ photochemistry to $HO_2$ sink
in this coastal region.
**3.3.2 Impact on Urban-nonurban Differences**

We further explore the reversed $O_3$ differences by separating the individual



contribution of climate and anthropogenic changes using the interpretable machine
learning model (**Figure 9**). The results demonstrate that atmospheric chemical
processes and meteorological conditions commonly dominate the urban-nonurban
discrepancies in $O_3$ levels. During the lockdown period, the diminished reduction in
boundary layer height and radiation flux across urban areas, compared to nonurban
areas in 2020, decelerated the expected decline of $O_3$ concentrations, leading to urban
$O_3$ levels exceeding those of nonurban areas (Figure S15). Concurrently, a narrowing
difference in urban and nonurban temperatures, despite an overall cooling from 2019
to 2020, favored $O_3$ formation in urban regions during the summertime. Additionally,
PM2.5 emerged as the principal anthropogenic factor inverting the urban-nonurban $O_3$
disparity over the course of 2019 to 2020. Its contribution to ozone shifted from being
lower in urban areas to exceeding that in nonurban areas, revealing that the decreased
reactive uptake of $HO_2$ from aerosols induced by a more substantial reduction in $PM_{2.5}$
in urban areas made the larger contribution to $O_3$ production (Ivatt et al., 2022; Li et al.,
2017). Meanwhile, the $O_3$ formation regimes also determine the response of $O_3$ to the
changes in its precursors. The abatement of $NO_x$ exhibited enhanced efficacy for $O_3$
mitigation in nonurban areas, which, in 2020, shifted predominantly toward a $NO_x$-
limited regime, in contrast to urban regions that remained constrained by a more VOC-
limited or oscillated between $NO_x$ and VOCs regimes (**Figure 8**b). **Figure 8**d shows
that urban regions, characterized by elevated $NO_x$ emission, exhibited a higher
proportion of VOC-limited, and the fraction of aerosol-inhibited areas increased from



2019 to 2020, resulting in the control benefits of urban $O_3$ pollution in 2020 are partially
offset by the nonlinear response of $O_3$ to a greater reduction in $NO_2$ and $PM_{2.5}$, and a
smaller decrease in HCHO relative to nonurban areas. Consequently, $O_3$ exhibits a
lower reduction in urban areas as a result of the aforementioned changes.

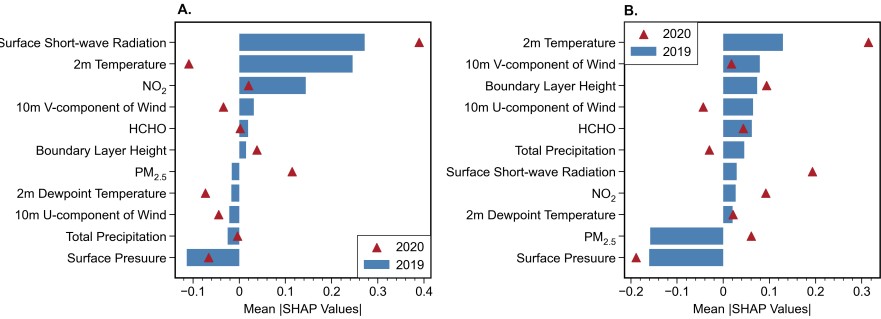


**Figure 9.** The changes of the mean absolute SHAP values disparities in urban-nonurban
from 2019 to 2020 across Shandong, China during the lockdown periods (A) and
summertime (B).
**4. CONCLUSIONS**

Based on the evaluation of the non-linearity of $O_3$-$NO_x$-VOCs-aerosols chemistry

captured by interpretable ML model based on spatially resolved multi-pollutants
estimations, this study assesses three various chemical regimes by tracking $NO_x$, VOCs
and aerosols with surface $NO_2$, HCHO and $PM_{2.5}$. We conclude that with the effective
reduction of $PM_{2.5}$ pollution, the sensitivity of $O_3$ to VOCs will increase, necessitating
that government agencies further intensify the regulation of VOC emissions. In the
Jiaodong Peninsula of Shandong Province, coastal areas with relatively minor primary
pollutants are widely found to be under an aerosol suppression regime, illustrating that
ozone regime inference based on machine learning can serve as an alternative to



determining the aerosol suppression regime through the rate of radical termination in
atmospheric chemical models. The $O_3$ regime in other areas of Shandong generally
transited from the VOC-sensitive regime in urban to a more $NO_x$-sensitive regime in
nonurban. We estimate that the substantial anthropogenic emission reduction of $PM_{2.5}$
and $NO_2$ is the main anthropogenic driver of the reversal of traditional urban-nonurban
discrepancy in $O_3$ levels. This shift underlines the intricate balance between emission
reduction and ozone formation mechanisms, suggesting that nuanced understanding
and targeted interventions are necessary to manage and mitigate the health and
environmental impacts of such disparities. To preclude exacerbated $O_3$ pollution
resulting from the shift of many regions from $NO_x$-limited to VOC-limited regimes and
the decline in heterogeneous $HO_2$ uptake induced by $PM_{2.5}$ reduction in urban areas,
emission policies aimed at decreasing $NO_x$ to reduce $O_3$ levels would only be effective
with stringent VOC emission abatement when $PM_{2.5}$ is concurrently decreased.

Ozone formation is highly nonlinear, so accurate estimations are essential to infer

its chemical regimes. The evaluation of model performance indicates that it can be
readily extended to any other domain thanks to the unified architecture. Anyone can
easily utilize the model to estimate ground-level pollutants that intelligently consider
spatial-temporal neighborhood information based on their customized input data. Our
model further improved the spatial resolution to sub-km using TROPOMI and MODIS
retrievals via spatiotemporal autocorrelation downscaling of AiT. The "black box" AiT
can be more physically interpretable by SHAP, enabling the evaluation of the



594 significance of each input variable (Figure S16). The season trends show the highest

595 contribution, followed by emission proxies and meteorological conditions. The

596 approach leads to these potentially surprising results that bring clarity to the growing

597 space of methods.

598  Although our study endeavors to establish $O_3$ formation regimes involving $NO_x$,

599 VOCs and aerosols, and the method identifies an aerosol inhibited from a statistical

600 perspective, it is subject to certain uncertainties to rely on the relatively poor data

601 quality of HCHO and the unsegregated multiple impacts of aerosols, such as $N_2O_5$

602 uptake, $NO_2$ uptake, $HO_2$ uptake and light extinction (Tan et al., 2022). We have made

603 efforts to integrate all required surface pollutant concentrations into a unified model,

604 while the absence of ground-level HCHO monitoring data compelled us to tap into an

605 alternative methodology. The retrieval error of surface HCHO and the system error

606 between its retrieval approach and the AiT model degrade the ability of ML to identify

607 the $O_3$ sensitivity. Meanwhile, the notion of ozone regimes is only appreciated in

608 photochemically active environments where the $RO_x$-$HO_x$ cycle is active (Souri et al.,

609 2023). The definition of $NO_x$-limited or VOC-limited is meaningless in nighttime

610 chemistry, where $NO$-$O_3$-$NO_2$ partitioning is the primary driver. The surface daytime

611 pollutant estimations with finer resolutions in space and time based on a unified

612 modeling framework will offer an unprecedented view to characterize the near-surface

613 $O_3$ formation regimes.



## Competing Interests

The authors declare that they have no conflict of interest.

## Acknowledgments

The work was financially supported by the National Natural Science Foundation
of China (project No. 22236004) and Taishan Scholars (No. ts201712003).

## Code and Data Availability

The Air Transformer deep learning framework is available on GitHub
(https://github.com/myles-tcl/Air-Transformer), which provides the scripts for
spatiotemporal data extraction, normalization, model training, and estimating of multi-
pollutants. The sources of input data in the Air Transformer can be found in Table S1.
The estimation of the Air Transformer can be downloaded from Zenodo:
https://zenodo.org/records/10071408 (Tao, 2023).

## Author Contributions

CT: Methodology, Software, Validation, Formal analysis, Investigation, Data
Curation, Writing-Original Draft, Visualization. YP: Conceptualization, Writing-
Review & Editing. QZ: Writing-Review & Editing, Project administration, Funding
acquisition. YZ: Methodology, Writing-Review & Editing. BG: Software, Writing-
Review & Editing. QW: Supervision, Writing-Review & Editing. WW: Supervision,
Writing-Review & Editing.



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
