# Peer review of "Diagnosing Ozone-NOx-VOCs-Aerosols Sensitivity to Uncover Urban-nonurban Discrepancies in Shandong, China using Transformer-based High-resolution Air"

_EGUsphere, 2023_

## Author Comment (AC1)

**Response to Comments on the Manuscript (egusphere-2023-2640)**

**Diagnosing Ozone-NO$_x$-VOCs-Aerosols Sensitivity to Uncover Urban-nonurban Discrepancies in Shandong, China using Transformer-based High-resolution Air Pollution Estimations**

Dear Referee,

We are grateful to the reviewer for their time and energy in providing helpful comments and suggestions which have significantly improved the manuscript. We have revised our manuscript according to all of the reviewer's comments to address these concerns in full.

The referee's comments and concerns are answered in detail point-by-point. The referee's comments are shown in black and the authors' responses are shown in blue.

**Reviewer #1 Comments to Author:**

The authors developed a deep learning framework to estimate surface O$_3$, NO$_2$, and PM$_{2.5}$ concentrations, and investigated urban-nonurban difference and ozone-NO$_x$-VOCs-aerosols sensitivity for ozone pollution in Shandong. This manuscript needs to be revised before it can be published.

1. Theme: There are two logics based on the title, abstract, conclusion, and the last paragraph of the introduction. One theme is "High Resolution Air Pollution Estimation", while the spatial characterization of pollution and urban-rural differences are further investigated in order to illustrate the value of the application of this deep learning framework. The other theme is to study the spatial characteristics of pollution in Shandong, and a deep learning approach is used. Authors should consider the perspective of the writing.

**Response:** We appreciate the reviewer's insight into the thematic presentation of our work. In response, we have carefully revised our manuscript to unify the

themes of air pollution estimation and the spatial characteristics of pollution. We clarified our narrative to emphasize how our deep learning framework not only advances the estimation of air pollutants but also provides valuable insights into urban-nonurban differences and ozone dynamics, and further explain the logical progressive relationship between air pollutant estimation, urban-nonurban differences, and ozone sensitivity analysis. The revision aims to present a cohesive narrative that aligns with both the methodological advancements and their application in environmental analysis. Our revision emphasizes the theme of urban-nonurban disparities in ozone, wherein we estimate multiple pollutant concentrations to analyze these differences as well as the impact of precursor emissions on it. Subsequently, through interpretable machine learning, we infer the ozone photochemical regime to unveil the influence of meteorology and chemical factors on ozone disparities.

The main revision in the title is as follows in lines 1-3: "Diagnosing Ozone-$NO_x$-VOCs-Aerosols Sensitivity and Uncovering Causes of Urban-Nonurban Discrepancies in Shandong, China using Transformer-Based Estimations".

The main revision in the abstract is as follows in lines 22-43: "Narrowing surface ozone disparities between urban and nonurban areas escalate health risks in densely populated urban zones. A comprehensive understanding of the impact of ozone photochemistry on this transition remains constrained by current knowledge of aerosol effects and the availability of surface monitoring. Here we reconstructed spatiotemporal gapless air quality concentrations by a novel Transformer deep learning (DL) framework capable of perceiving spatiotemporal dynamics to analyze ozone urban-nonurban differences. Subsequently, the photochemical effect on these discrepancies was analyzed by elucidating shifts in ozone regimes using an interpretable machine learning method. The evaluations of DL model exhibited average out-of-sample cross-validation coefficient of determination of 0.96, 0.92, and 0.95 for ozone, nitrogen dioxide, and fine particulate matter ($PM_{2.5}$), respectively. The ozone sensitivity in nonurban areas, dominated by nitrogen oxide ($NO_x$)-limited regime, was observed to shift towards

increased sensitivity to volatile organic compounds (VOCs) when extended to urban areas. A third 'aerosol-inhibited' regime was identified in the Jiaodong Peninsula, where the uptake of hydroperoxyl radicals onto aerosols suppressed ozone production under low $NO_x$ levels during summertime. The reduction of $PM_{2.5}$ would increase the sensitivity of ozone to VOCs, necessitating more stringent VOC emission abatement for urban ozone mitigation. In 2020, urban ozone levels in Shandong surpassed those in non-urban, primarily due to a more pronounced decrease in the latter resulting from stronger aerosol suppression effects and lesser $PM_{2.5}$ reductions. This case study demonstrates the critical need for advanced spatially resolved models and interpretable analysis in tackling ozone pollution challenges."

The main revision in the introduction is as follows in lines 105-122: "In this study, we aim to analyze the evolving dynamics of urban-nonurban $O_3$ differences between 2019 and 2020. The roles of emission discrepancies and nonlinearity of $O_3$-$NO_x$-VOCs-aerosols photochemical processes in shaping these $O_3$ variations were deeply dissected. To achieve a comprehensive analysis, we employed a new spatiotemporal Transformer framework that paid special attention to air mass transport and dispersion affected by the spatial-temporal correlations, to reconstruct the spatially gapless air quality datasets based on satellite data, ground-level observations, and meteorological reanalysis. The estimations are particularly vital for regions lacking dense ground-based monitors, ensuring that our understanding of $O_3$ dynamics in urban-nonurban areas and formation regimes is not limited by geographical constraints in data availability. Surface $O_3$ formation regimes in Shandong province were inferred by the classic XGBoost model coupled with Shapley Additive exPlanations (SHAP), which identifies the impact of meteorological conditions and photochemical indicators (i.e. $PM_{2.5}$ as a proxy for aerosols, $NO_2$ as a proxy for $NO_x$, and HCHO as a proxy for VOCs) on $O_3$. The innovative Transformer-based modeling and interpretable machine learning analysis approaches are expected to enable new applications such as those of air quality simulation and $O_3$ formation regimes studies."

The main revision in conclusions is as follows in lines 618-628: "The purpose of the current study was to diagnose the non-linearity of $O_3$-$NO_x$-VOCs-aerosols chemistry using an interpretable ML model based on spatially resolved multi-pollutant estimations for determining the causes of changing differences in $O_3$ levels between urban and non-urban areas. Our study represents the first attempt to develop an advanced DL model that reconstructs the concentrations of multiple pollutants and subsequently infers the aerosol-inhibited regime from observations. This innovative approach provides further support for investigating the impact of precursor emissions and aerosol on the urban-nonurban differences in $O_3$ levels.". More detailed modifications can be found in the manuscripts with tracking changes.

2. The study is divided into two main parts: one on estimating ozone concentrations and studying urban-rural differences, and the other on ozone sensitivity. The logical relationship between the two parts is not very coherent. Ozone sensitivity does not adequately explain the variations and differences in ozone concentrations between urban and rural areas and in different years. In other words, if ozone concentrations are not estimated, it does not seem to affect the results of the ozone sensitivity study.

**Response:** We appreciate the insightful feedback regarding the perceived coherence between the two parts of our study. The initial design aimed to comprehensively understand the multifaceted nature of ozone, especially for uncovering the urban-rural differences. However, we recognized the need for a more explicit linkage to better articulate the study's coherence. The variations in ozone levels between urban and nonurban areas are influenced by a complex interplay of factors, including emission sources (precursors and $PM_{2.5}$), atmospheric chemistry (ozone photochemical sensitivity), and meteorological conditions. Thus, we first discussed the impact of the distribution pattern of $NO_2$, HCHO and $PM_{2.5}$ on ozone urban-nonurban disparities. And then the sensitivity of ozone to these pollutants was inferred to understand these differences.

Furthermore, we examined the influence of meteorological conditions. The high-resolution estimation of air quality concentration lays the groundwork for the above analysis by providing a comprehensive dataset, which compensates for potential biases caused by the sparse spatial distribution of monitoring sites, including site imbalance between urban and nonurban areas in the analysis of urban-rural differences and a limited number of site used to determine thresholds in the $HCHO/NO_2$ ratio method. Our intention is that by reconstructing surface air quality concentration, we can accurately assess ozone formation regimes and urban-nonurban differences and then uncover the cause of these differences. We have modified the text to strengthen the logical connection and ensuring the relevance of each part. Considering that ozone sensitivity alone cannot fully account for urban-nonurban differences, we have revised the title to "Diagnosing Ozone-$NO_x$-VOCs-Aerosols Sensitivity and Uncovering Causes of Urban-Nonurban Discrepancies in Shandong, China using Transformer-Based Estimations" for more accurately summarize the content of the manuscript. Moreover, the estimation of ozone concentrations is crucial for inferring ozone sensitivity. The novel machine-learning-based method supplies the analysis of aerosol-inhibited regime based on observation. The inference in the atmospheric chemical transportation model is highly dependent on emission inventories and prior chemical mechanisms. we enhanced the study's overall coherence and impact by more explicitly connecting ozone concentration estimates with sensitivity analysis in the revised manuscript.

3. Ozone formation regimes: From Fig.8, the $NO_x$-limited regime dominates, especially Fig.8D shows that the proportion of $NO_x$-limited is almost 1.0, which is not quite consistent with the authors' conclusions.

**Response:** Thanks for your insightful comment. We sincerely apologize for any confusion caused by the inaccuracies in our previously stated conclusions regarding the ozone formation regimes. Upon closer examination, we acknowledge that our initial interpretation, suggesting an increased sensitivity of ozone to $NO_x$ transitioning from urban to non-urban areas and erroneously

concluding urban areas to be predominantly VOC-limited. In response to your comment, we have conducted a thorough reanalysis of the pertinent data and have updated our findings accordingly in Figure 8. This revised analysis provides a more precise and quantified insight into the distribution of ozone formation regimes across different urban and non-urban settings. Specifically, our updated results indicate that in certain cities, the prevalence of the VOC-limited regime within urban areas varies between 15% to 43%. This contrasts with non-urban areas, where the VOC-limited regime is significantly less common.

The main revision in the abstract is as follows in lines 32-35: "The ozone sensitivity in nonurban areas, dominated by nitrogen oxide ($NO_x$)-limited regime, was observed to shift towards increased sensitivity to volatile organic compounds (VOCs) when extended to urban areas."

The main revision in section 3.3.1 is as follows in lines 536-639: "Moving along an urban-to-rural gradient, reactions dominated by $RO_x$ radical self-reactions are continuously enhanced with increasing $NO_x$ SHAP values, resulting in the majority of rural Shandong being situated in $NO_x$-limited regimes.", and lines 555-572: "In several cities, including Binzhou, Zibo, Liaocheng, Linyi, and Jining, a greater proportion of urban areas, as compared to their nonurban counterparts, exhibited a VOC-limited regime in 2019, as indicated by the prevalence of red regions in Figure 8D. The percentage of urban areas in these cities under a VOC-limited regime ranges from 15% to 43%, in stark contrast to non-urban areas where such a regime is typically rare (Figure 8F). The comparison of $O_3$ sensitivities from 2019 to 2020 shows a regional shift towards increased sensitivity to aerosol and $NO_x$, along with a decreased VOC sensitivity as a result of $NO_x$ reduction (Figure 8A-C). This shift has led to the majority of areas in Shandong being dominated by a $NO_x$-limited regime in 2020, with an expanded aerosol-inhibited regime region in the Jiaodong Peninsula (Figure 8E). Additionally, the discrepancy in $O_3$ formation sensitivity between urban and non-urban areas has been diminishing during this period (Figure 8C). As illustrated in Figure 9, while the ozone regime transitions towards $NO_x$-limited, there is a marked shift towards greater aerosol sensitivity

across nearly 90% of areas, leading to a 1.6% increase in aerosol-inhibited grids. Compared to nonurban regions, a higher number of grids in urban areas demonstrate a shift towards $NO_x$ sensitivity. Conversely, urban areas that were predominantly aerosol-inhibited in 2019 showed a lower sensitivity shift towards $NO_x$."

[Figure]

**Figure 8.** Comparison of geographical distribution for ozone formation regimes between 2019 and 2020 in the summertime. All surface daily $O_3$, $PM_{2.5}$, and $NO_2$ estimations from Air Transformer (AiT) are averaged over each month from May to October 2019-2020 for matching monthly HCHO derived from TROPOMI (500 * 500 m). (A, B) Geographical distribution of fractional contribution of chemical factors representing $O_3$ formation regimes. The ternary phase diagram in the legend depicts the normalized fraction of SHAP values for $O_3$ attributed to HCHO, $NO_2$, and $PM_{2.5}$ at the surface, representing VOC-limited (red), aerosol-inhibited (green), and $NO_x$-limited (blue) regimes, respectively. (C) Statistical Changes in the fractional contribution of chemical factors. (D, E) Geographical distribution of $O_3$ chemical regimes. (F) Proportion of three $O_3$ chemical regimes across urban and nonurban areas in 2019 in Shandong (SD), and individual cities (BZ: Binzhou, ZB: Zibo, LC: Liaocheng, LY: Linyi, JNI: Jining).

[Figure]

**Figure 9.** Geographical distribution of changes in ozone sensitivity from 2019 to 2020 in summertime (A). Comparison of ozone sensitivity changes across areas dominated by different chemical regimes in 2019 between urban and non-urban areas (B).

4. Specification of figures: Do Figures 7A and B share a colorbar to indicate $O_3$ concentration? In fact, Figure 7B contains the information from Figure 7A. In Figure 7D, what does the arrow next to $PM_{2.5}$ mean? Figure 8C is a legend for Figures 8A and B, making it difficult to understand. In Figure 9, it is not appropriate to represent one variable in dots and one in columns as they are of the same kind.

**Response:** We deeply appreciate your constructive comments and have taken the following steps to address the concerns raised:

(1) Both Figures 7A and B utilize a shared color bar to indicate $O_3$ concentrations, enhancing comparability. We have now included a detailed explanation in the figure captions to clarify this.

(2) While it is true that Figure 7B encompasses the data presented in Figure 7A, we maintain Figure 7A to provide a more intuitive comparison with previous studies and to affirm the robustness of our dataset analysis. The classic $O_3$-VOC-

NO$_x$ isopleths presented in Figure 7A offer a direct and easily interpretable visual representation, hence our decision to retain it for its comparative value and reliability verification.

(3) The arrow adjacent to PM$_{2.5}$ originally intended to signify an increase in PM$_{2.5}$ concentrations. To clarify, we have replaced the arrow with the term "increasing" to directly convey the intended meaning without ambiguity.

(4) Acknowledging the difficulty in interpreting Figure 8C as a standalone legend for Figures 8A and B, we have incorporated the legend directly into Figures 8A and B.

(5) Upon reflection, we agree that representing variables of the same nature in different graphical forms (dots and columns) in Figure 10 could potentially confuse the reader. We have thus revised Figure 10, opting for a consistent bar graph representation for both sets of data.

[Figure]

**Figure 7.** (A) O$_3$ concentrations as a function of surface HCHO and NO$_2$. (B) O$_3$ concentrations as a function of surface HCHO, NO$_2$, and PM$_{2.5}$. Both A and B utilize a shared color bar to indicate O$_3$ concentrations, enhancing comparability. (C) Relationship between O$_3$, and NO$_2$, HCHO, and surface short-wave radiation flux. The paired O$_3$, HCHO, NO$_2$, and solar radiation are divided into 100 bins based on PM$_{2.5}$ and then the averaged concentrations (y-axis) are calculated for each PM$_{2.5}$

bin (x-axis). (D) Changes in HCHO/NO$_2$-O$_3$ relationship in response to changing PM$_{2.5}$ by XGBoost model. The solid lines are fitted with four-order polynomial curves, and the shading indicates 95% confidence intervals. (E-F) The interaction SHAP values reveal an interesting hidden relationship between pairwise variables (PM$_{2.5}$ and NO$_2$, HCHO) and O$_3$.

[Figure]

**Figure 10.** Comparison of urban-nonurban disparities in meteorological conditions (A), and mean absolute SHAP values (B) between 2019 and 2020 across Shandong, China during the summertime.

[Figure]

**Figure S19.** Comparison of urban-nonurban disparities in meteorological conditions (A), and mean absolute SHAP values (B) between 2019 and 2020 across Shandong, China during the COVID period.

---

## Author Comment (AC2)

**Response to Comments on the Manuscript (egusphere-2023-2640)**

**Diagnosing Ozone-NO$_x$-VOCs-Aerosols Sensitivity to Uncover Urban-nonurban Discrepancies in Shandong, China using Transformer-based High-resolution Air Pollution Estimations**

Dear Referee,

We are grateful to the reviewer for their time and energy in providing helpful comments and suggestions which have significantly improved the manuscript. We have revised our manuscript according to all of the reviewer's comments to address these concerns in full.

The referee's comments and concerns are answered in detail point-by-point. The referee's comments are shown in black and the authors' responses are shown in blue.

**Reviewer #2 Comments to Author:**

This study developed a novel spatiotemporal deep learning model for concurrent prediction of three air pollutants (ozone, NO$_2$, PM$_{2.5}$). The authors used the generated fine-scale concentrations to assess urban-nonurban differences and ozone-NO$_x$-VOCs-aerosols sensitivity in Shandong, China. To facilitate the analysis, interpretable machine learning was employed to handle nonlinearity and isolate impacts of drivers relating to ozone photochemistry. The methodology is solid, and the findings are important for the development of ozone control strategies, though a few issues remain.

1. Line 259: Please explain the possible reason why NO$_2$ has significantly lower out-of-site CV-R$^2$ (0.75) than ozone and PM$_{2.5}$ (>0.9), note that the out-of-sample CV results are comparable across all pollutants?

**Response:** The decreased $R^2$ for $NO_2$ in out-of-site cross-validation could result from the short atmospheric chemistry lifetime of $NO_2$, which leads to greater potential disparities in the relationship of satellite column density and surface $NO_2$ between various monitoring stations. Meanwhile, previous studies also show the same problem. For example, Wei et al. (Wei et al., 2022) estimate the ground-level $NO_2$ surveillance with an average out-of-city (out-of-sample) cross-validation $R^2$ of 0.71 (0.93) using interpretable spatiotemporally weighted artificial intelligence. The same trouble of the underestimation of high values leads to the reduced evaluation metric.

References:

Wei, J., Liu, S., Li, Z., Liu, C., Qin, K., Liu, X., Pinker, R.T., Dickerson, R.R., Lin, J., Boersma, K.F., Sun, L., Li, R., Xue, W., Cui, Y., Zhang, C., Wang, J., 2022. Ground-Level $NO_2$ Surveillance from Space Across China for High Resolution Using Interpretable Spatiotemporally Weighted Artificial Intelligence. Environ. Sci. Technol. acs.est.2c03834. https://doi.org/10.1021/acs.est.2c03834

2. Lines 265-266: In evaluating stability and robustness of the model, it would be interesting to see if the CNEMC-trained model can obtain local concentration variations and interpretation outcomes similar to that from the CNEMC+SDEM-trained model.

**Response:** Thank you for your insightful comments. We have conducted the suggested comparison and incorporated the results into Figure 3, comparing multiple datasets. We apologize for the initial error in the verification of the CNEMC-trained model on the SDEM dataset, which has also now been rectified. Additionally, we have included a kernel density estimation result to visually represent these verification results. It is observed that the CNEMC-trained model exhibits only an acceptable degradation in predictive accuracy on the SDEM dataset compared to out-of-site cross-validation of AiT (Figure S6). The outcome also reveals a similar spatial gradient at the urban scale (Figure 3). Meanwhile, the

comparison results on a daily scale during sandstorms show that although the model trained with CNEMC data exhibits some overestimation or underestimation in certain areas, it demonstrates similar spatial distribution and temporal variation trends as the model trained with all data (Figure S9). These results reveal the reliability of my deep learning model and the promising prospect of continuously improving the model's generalization ability with more ground-level monitoring data.

The main revision is as follows in lines 294-296: "This spatial gradient is also captured by AiT trained with CNEMC data, revealing the reliability of the deep learning model structure", lines 327-329: "The model trained solely on CNEMC data is also capable of effectively capturing the drastic changes in air quality during the pollution episode (Figure S9)", and lines 637-641: "Meanwhile, the results between AiT trained with all data and that trained exclusively with CNEMC data across various spatiotemporal scales underscore the promising prospect for improving the model's generalization ability with more ground-level monitoring data and the growing space of methods.".

[Figure]

**Figure 3.** Spatial distribution of the annual mean (A-E) $O_3$, (K-O) $NO_2$, and (U-Y) $PM_{2.5}$ concentrations from observations, Air Transformer (AiT), CNEMC-trained

AiT, Random Forest (RF) and ChinaHighAirPollutants (CHAP), respectively, in 2019. The region enclosed by the red rectangular box corresponds to the zoomed-in maps of the satellite (© Tianditu: www.tianditu.gov.cn) and pollutant concentrations at a city scale for the capital city of Shandong Province, Jinan.

[Figure]

**Figure S6**. Validation for daily ground-level $O_3$, $NO_2$, and $PM_{2.5}$ concentration in the SDEM dataset based on the AiT model trained by monitoring data of CNEMC.

[Figure]

**Figure S9**. Comparison of spatial distribution between estimations from AiT trained with all data and AiT with CNEMC data during the dust storm.

3. Lines 322-323: The time span of the training data should be given, as that information is important to understand whether the good agreements between measurements and estimations reflect fitting or prediction performance.

**Response:** Thanks for your kind suggestions. We have added the time information of the training dataset in section 2.2 at lines 197-200: "The aggregated feature data from June 2019 to June 2021 were utilized to train and validate the model through cross-validation (CV), where the optimal model, trained based on out-of-sample CV, was used to estimate multiple pollutant concentrations during the study period, which was then employed for subsequent analysis.".

4. How many monitoring stations are there in urban areas? A map highlighting the urban and nonurban areas is recommended for intuitive understanding.

**Response:** We counted the number of monitoring sites in urban and non-urban areas in Table S4. The number of urban sites in 13 cities exceeds that in non-urban areas. Particularly in cities like JNA, LC, LY, QD, and YT, the disparity in the number of urban and non-urban sites is significant, leading to urban-nonurban differences that are contrary to those observed in AiT. We also added the map of urban extents in supporting information as Figure S11.

The main revision for the number of monitoring stations is as follows in lines 397-400: "The notable disparity between the number of urban and non-urban sites in cities such as JNA, LC, LY, QD, and YT results in a pattern of urban-nonurban differences that contrasts markedly with the observed in AiT (Table S4).".

**Table S4.** The number of monitoring stations across urban and non-urban areas. (YT: Yantai, BZ: Binzhou, DY: Dongying, WH: Weihai, DZ: Dezhou, JNA: Jinan, QD: Qingdao, WF: Weifang, ZB: Zibo, LC: Liaocheng, LW: Laiwu, TA: Taian, LY: Linyi, RZ: Rizhao, JNI: Jining, HZ: Hezhe, ZZ: Zaozhuang)

| City Name | BZ | DY | DZ | HZ | JNA | JNI | LC | LW | LY |
|-----------|----|----|----|----|-----|-----|----|----|-----|
| Non-urban | 9 | 2 | 10 | 6 | 2 | 6 | 7 | 2 | 8 |
| Urban | 7 | 11 | 14 | 14 | 17 | 15 | 15 | 1 | 14 |
| City Name | QD | RZ | TA | WF | WH | YT | ZB | ZZ | |
| Non-urban | 1 | 5 | 4 | 9 | 3 | 3 | 10 | 2 | |
| Urban | 11 | 5 | 7 | 15 | 7 | 18 | 6 | 8 | |

The main revision for the map of urban extents is as follows in lines 373-374: "The urban extents in Shandong Province in 2019 are depicted in Figure S11."

[Figure]

**Figure S11**. Urban extents (red) in Shandong province, China in 2019.

5. There is a lack of validation for the XGBoost model, given that reliability of interpretation outcomes should be based on the model with high accuracy.

**Response:** Thanks for your insightful comments. We added the results of 10-fold cross-validation as shown in Figure S16.

The main revision is as follows in lines 500-502: "As depicted in Figure S16, the performance of the XGBoost model is robust, evidenced by a high $R^2$ value of 0.99 coupled with a low RMSE of 3.24 μg/m³ and MAE of 2.33 μg/m³".

[Figure]

**Figure S13.** Results of 10-fold cross-validation in validation dataset based on XGBoost for modeling the nonlinear response of monthly O₃ variations to

meteorology and chemical indicators from 2019 to 2020.

6. Please provide more explanations for the SHAP interaction values. The statement "lower NO$_2$ … could diminish the formation of ozone under high PM$_{2.5}$ concentrations" (Line 456) is difficult to follow. In Figure 7e, lower NO$_2$ and negative PM$_{2.5}$-NO$_2$ SHAP interaction values are observed at lower PM$_{2.5}$ levels.

**Response:** Thanks for the reviewer's suggestion. We have added more information in section 3.3 for easier understanding.

The main revision is as follows in lines 466-479: "The SHAP interaction plot in Figure 7e, f illustrates that the influence of NO$_2$ and HCHO on O$_3$ formation is not constant and is influenced by the levels of PM$_{2.5}$. Typically, at a certain level of PM$_{2.5}$, the lower NO$_2$ concentration, the stronger inhibition effect on O$_3$ production. This could be due to aerosols exerting stronger suppression through the HO$_2$ sink at lower NO$_x$ levels. As the concentration of PM$_{2.5}$ increases, often involving a concurrent increase in NO$_2$ as a key precursor, there is a greater need for higher levels of NO$_2$ to be converted into nitrous acid (HONO) through the heterogeneous uptake by aerosols. This process produces more OH radicals, which facilitate photochemical O$_3$ formation, offsetting the increased inhibitory effect of the HO$_2$ sink. Under high PM$_{2.5}$ concentrations, an increase in NO$_2$ along with a decrease in HCHO enhances their effect on the promotion of O$_3$ formation. This enhancement could be caused by increased titration of O$_3$ by NO, resulting from weaker conversion from NO to NO$_x$ through the RO$_x$ radical. Meanwhile, the impact of HCHO shifts from promotion to suppression as PM$_{2.5}$ pollution intensifies.".

7. Figure 8d shows that the NO$_x$-limited regime dominates in urban areas. Please confirm.

**Response:** Thanks for your insightful comment. We sincerely apologize for any confusion caused by the inaccuracies in our previously stated conclusions regarding the ozone formation regimes. Upon closer examination, we acknowledge that our initial interpretation, suggesting an increased sensitivity of

ozone to $NO_x$ transitioning from urban to non-urban areas and erroneously concluding urban areas to be predominantly VOC-limited. In response to your comment, we have conducted a thorough reanalysis of the pertinent data and have updated our findings accordingly in Figure 8. This revised analysis provides a more precise and quantified insight into the distribution of ozone formation regimes across different urban and non-urban settings. Specifically, our updated results indicate that in certain cities, the prevalence of the VOC-limited regime within urban areas varies between 15% to 43%. This contrasts with non-urban areas, where the VOC-limited regime is significantly less common.

The main revision in the abstract is as follows in lines 32-35: "The ozone sensitivity in nonurban areas, dominated by nitrogen oxide ($NO_x$)-limited regime, was observed to shift towards increased sensitivity to volatile organic compounds (VOCs) when extended to urban areas."

The main revision in section 3.3.1 is as follows in lines 536-639: "Moving along an urban-to-rural gradient, reactions dominated by $RO_x$ radical self-reactions are continuously enhanced with increasing $NO_x$ SHAP values, resulting in the majority of rural Shandong being situated in $NO_x$-limited regimes.", and lines 555-572: "In several cities, including Binzhou, Zibo, Liaocheng, Linyi, and Jining, a greater proportion of urban areas, as compared to their nonurban counterparts, exhibited a VOC-limited regime in 2019, as indicated by the prevalence of red regions in Figure 8D. The percentage of urban areas in these cities under a VOC-limited regime ranges from 15% to 43%, in stark contrast to non-urban areas where such a regime is typically rare (Figure 8F). The comparison of $O_3$ sensitivities from 2019 to 2020 shows a regional shift towards increased sensitivity to aerosol and $NO_x$, along with a decreased VOC sensitivity as a result of $NO_x$ reduction (Figure 8A-C). This shift has led to the majority of areas in Shandong being dominated by a $NO_x$-limited regime in 2020, with an expanded aerosol-inhibited regime region in the Jiaodong Peninsula (Figure 8E). Additionally, the discrepancy in $O_3$ formation sensitivity between urban and non-urban areas has been diminishing during this period (Figure 8C). As illustrated in Figure 9, while the ozone regime transitions

towards NO$_x$-limited, there is a marked shift towards greater aerosol sensitivity across nearly 90% of areas, leading to a 1.6% increase in aerosol-inhibited grids. Compared to nonurban regions, a higher number of grids in urban areas demonstrate a shift towards NO$_x$ sensitivity. Conversely, urban areas that were predominantly aerosol-inhibited in 2019 showed a lower sensitivity shift towards NO$_x$."

[Figure]

**Figure 8.** Comparison of geographical distribution for ozone formation regimes between 2019 and 2020 in the summertime. All surface daily O$_3$, PM$_{2.5}$, and NO$_2$ estimations from Air Transformer (AiT) are averaged over each month from May to October 2019-2020 for matching monthly HCHO derived from TROPOMI (500 * 500 m). (A, B) Geographical distribution of fractional contribution of chemical factors representing O$_3$ formation regimes. The ternary phase diagram in the legend depicts the normalized fraction of SHAP values for O$_3$ attributed to HCHO, NO$_2$, and PM$_{2.5}$ at the surface, representing VOC-limited (red), aerosol-inhibited (green), and NO$_x$-limited (blue) regimes, respectively. (C) Statistical Changes in the fractional contribution of chemical factors. (D, E) Geographical distribution of O$_3$ chemical regimes. (F) Proportion of three O$_3$ chemical regimes across urban and nonurban areas in 2019 in Shandong (SD), and individual cities (BZ: Binzhou, ZB: Zibo, LC: Liaocheng, LY: Linyi, JNI: Jining).

[Figure]

**Figure 9.** Geographical distribution of changes in ozone sensitivity from 2019 to 2020 in summertime (A). Comparison of ozone sensitivity changes across areas dominated by different chemical regimes in 2019 between urban and non-urban areas (B).

8. Minor typos and grammar errors need to be corrected. For example, Line 355: the upper right area of E, M, and U; Line 370: are shown in Figure 5, etc.

**Response:** Thank you for pointing out these typos and grammar errors. We have carefully modified them in the manuscript. We also conducted a thorough review of the entire text to ensure the accuracy and clarity.